# Role of Epidermal Growth Factor Receptor-Specific CAR-T Cells in the Suppression of Esophageal Squamous Cell Carcinoma

**DOI:** 10.3390/cancers14246021

**Published:** 2022-12-07

**Authors:** Chen Cheng, Heyang Cui, Huijuan Liu, Yueguang Wu, Ning Ding, Yongjia Weng, Weimin Zhang, Yongping Cui

**Affiliations:** 1Shenzhen Key Laboratory of Gastrointestinal Cancer Translational Research, Cancer Institute, Peking University Shenzhen Hospital, Shenzhen Peking University-the Hong Kong University of Science and Technology Medical Center, Shenzhen 518028, China; 2Shenzhen Bay Laboratory, Institute of Cancer Research, Shenzhen 518028, China; 3Key Laboratory of Carcinogenesis and Translational Research (Ministry of Education), Department of Molecular Oncology, Peking University Cancer Hospital and Institute, Beijing 100142, China

**Keywords:** esophageal squamous cell carcinoma (ESCC), cellular immunotherapy, chimeric antigen receptor-T (CAR-T), epidermal growth factor receptor (EGFR)

## Abstract

**Simple Summary:**

Esophageal squamous cell carcinoma (ESCC) is a high-incidence cancer in China for which treatment strategies and therapeutic effects are limited. In this study, we established cellular immunotherapy for ESCC using epidermal growth factor receptor (EGFR)-targeting chimeric antigen receptor (CAR)-expressing T cells. The goal of this project was to provide new and effective therapies for ESCC.

**Abstract:**

ESCC is a highly malignant tumor, and its morbidity and mortality in China account for more than 50% of the world’s total rates. As effective treatments are lacking, the 5-year survival rate of patients does not exceed 30%. CAR-T-cell-based immunotherapy has emerged as one of the most promising cancer treatments; however, there are relatively fewer reports regarding its application for ESCC. In this study, we conducted large-sample whole-genome sequencing (WGS) and RNA-seq analysis of patients with ESCC from China to examine the feasibility of EGFR-targeting CAR-T cells in the treatment of ESCC. We found much higher levels of EGFR gene amplification and overexpression in tumors than in the normal tissues, indicating that EGFR could be a promising target of CAR-T-cell-based immunotherapy in ESCC. Therefore, we tested EGFR-targeting CAR-T cells for lytic activity against ESCC cells as a model to establish cellular immunotherapy for ESCC. Five types of CAR-T cells targeting EGFR were constructed, two of which, CAR1-T and CAR2-T, showed a strong cytotoxicity against ESCC in in vitro and in vivo experiments. The results of this study suggest that CAR1-T and CAR2-T have the potential to be used for anti-ESCC immunotherapy in clinics.

## 1. Introduction

Esophageal carcinoma (EC) is a type of malignant neoplasm of the digestive tract originating from the esophageal epithelium. EC is one of the most common cancers in China, with obvious regional characteristics, and it is predicted that approximately 324,422 people could be newly diagnosed with EC and 301,135 people could die from the disease in China [1]. The principal histological types of EC are esophageal squamous cell carcinoma (ESCC) and esophageal adenocarcinoma (EAC). In China, more than 90% of patients with esophageal cancer have ESCC. Patients with early-stage ESCC lack specific symptoms, and thus are mostly diagnosed at the middle or late stages. Few therapeutic options and a poor prognosis of ESCC account for the low 5-year survival rate of patients, which is ~30% [2].

Monoclonal antibodies targeting programmed death-1 (PD-1) or PD-ligand 1 (PD-L1) are representative immune checkpoint inhibitors that have shown good therapeutic effects for EC in clinical trials [3,4,5]. However, these immunotherapeutic approaches produce durable responses in only a subset of patients with ESCC, many of which have primary or acquired resistance [6,7]. The mechanisms of resistance to PD-1/PD-L1 inhibitors include the loss of first and co-stimulation signals, major histocompatibility complex dysfunction, irreversible T cell exhaustion, and an immunosuppressive tumor microenvironment [7,8,9,10]. Thus, there is still a large demand for effective ESCC therapies, and their development is the focus of basic and clinical research.

In the past few years, immunotherapy with chimeric antigen receptor (CAR)-expressing T cells has provided significant positive effects for the treatment of hematologic malignancies and has emerged as one of the most promising therapeutic methods in cancer [11,12,13,14]. Therefore, the exploration of CAR-T cell immunotherapy application to ESCC may provide new avenues for the development of effective treatment approaches for this disease.

Our ESCC whole genome sequencing (WGS) and RNA sequencing (RNA-Seq) data revealed that epidermal growth factor receptor (EGFR) is one of the most amplification genes and it is highly expressed in ESCC, and it has low or no expression in normal mucosal tissues [15]. Previous studies have also indicated that ~80% [16,17,18,19] of patients with ESCC show high levels of EGFR expression. Therefore, we speculated that EGFR could be an excellent target for CAR-T cell therapy in ESCC. EGFR is overexpressed in various epithelial tumors, and its abnormal activation is closely correlated with cancer occurrence and development [20,21]. According to the literature, CAR-T cells targeting EGFR have been explored in non-small cell lung [22,23] and biliary tract [24], triple negative breast cancers [25,26], and glioblastoma [27,28], and some of these results have already been tested in clinical trials. Thus, Feng et al. used EGFR-targeting CAR-T cells to treat patients with advanced and refractory non-small cell lung cancer with a strong positivity for EGFR expression (more than 50%), and the results indicate that 7 of the 11 enrolled patients were evaluable, including two with significant tumor reduction and five with stable disease [23]. In another phase I clinical trial, EGFR-targeting CAR-T cells were used for patients with recurrent/metastasizing biliary tract carcinoma with a strong EGFR expression, and the findings indicated that among the 17 evaluable patients (of the total 19), 1 had 22 months of complete remission and 10 achieved stable disease [24]. Overall, these results indicate that anti-EGFR CAR-T-cell-based therapy may have a high clinical potential in different cancer types.

The aim of this study was to test the feasibility of EGFR-targeting CAR-T cells in the treatment of ESCC and to provide a theoretical basis for preclinical research. Our ESCC whole-genome sequencing (WGS) and RNA-seq data revealed high levels of EGFR amplification and expression in ESCC compared with paracancerous tissues. To target EGFR-expressing ESCC cells, we used the second-generation CAR structure to constructe anti-EGFR CAR in this study because it has been reported that the second-generation CAR structure has a better antitumor effect [29]. We constructed five anti-EGFR CARs with a single chain fragment variable (scFv) derived from EGFR antibodies and used them to transfect T lymphocytes, which were then compared for their in vitro and in vivo anticancer activities. The results showed that CAR1-T, CAR2-T, and CAR4-T cells had a high cytotoxicity against ESCC in vitro; among them, CAR1-T and CAR2-T were also able to clear ESCC in vivo. The successful preparation and functional identification of anti-EGFR CAR-T cells targeting ESCC will lay the foundation for using CAR-T cell-based immunotherapy to treat ESCC.

## 2. Materials and Methods

### 2.1. Western Blotting

ESCC cells were lysed on ice for 30 min using a RIPA buffer supplemented with a Protease Inhibitor Cocktail. After measuring the total protein concentration, 50 μg of protein was subjected to SDS-PAGE (5% stacking/8% separating gels) and then transferred to nitrocellulose membranes, which were blocked with 2% BSA for 2 h at 16–24 °C, and then incubated with an anti-EGFR antibody (Cell Signaling Technology, Danvers, MA, USA) overnight at 4 °C. The membranes were then incubated with horseradish peroxidase-labeled secondary antibodies (Jackson ImmunoResearch, West Grove, PA, USA), and proteins of interest were detected using chemiluminescence reagents (ThermoFisher, Waltham, MA, USA). Glyceraldehyde-3-phosphate dehydrogenase (GAPDH) (Proteintech, Wuhan, China) was used as the loading control.

### 2.2. Cell Lines

HEK293T cells were kindly gifted from Dr. H. Wang and human ESCC cell lines KYSE30, KYSE150, KYSE180, KYSE450, and TE1 were kindly supplied by Dr. Y. Shimada. HEK293T and ESCC cells were cultured in DMEM (GIBCO) and RPMI 1640 (GIBCO) medium, respectively, supplemented with 10% (*v*/*v*) fetal bovine serum (FBS) and 100 U/mL penicillin/streptomycin (GIBCO). All of the cells were free of mycoplasma and maintained at 37 °C and 5% CO_2_.

### 2.3. Lentivirus Production

HEK293T cells were seeded at 7.0 × 10^6^ cells per 10 cm dish, cultured for 15–18 h, and transfected at ~90% confluence using Lipo3000 (ThermoFisher, Waltham, MA, USA, L3000015) according to the manufacturer’s protocol. Virus-containing supernatants were collected 48 and 72 h post-transfection and filtered. Lentiviral particles were concentrated by centrifugation at 4000 rcf for 1 h at −4 °C in virus concentration tubes (Merck Millipore Ultracel-100 K, UFC910096) and were stored at −80 °C.

### 2.4. Human CD3^+^ T Cell Enrichment, Activation, and Multiplication

hPBMCs were obtained from human peripheral blood provided by Shanghai Liquan Hospital and hPBMCs cell isolation service provided by Milestone Biotechnologies. CD3^+^ T cells were enriched according to the instructions of the T Cell Enrichment Kit (Stemcell Technologies, Vancouver, BC, Canada, 19051), activated with anti-CD3/anti-CD28 Dynabeads (ThermoFisher, Waltham, MA, USA, 11131D) added at a ratio of 2:1, and proliferated. The cells were maintained in X-VIVO15 medium (Lonza, Basel, Switzerland, 04-418Q) supplemented with 5% (*v*/*v*) inactivated FBS (GIBCO, Grand Island, NY, USA, 1914970), 100 U/mL penicillin/streptomycin (GIBCO, Grand Island, NY, USA, 15140-122), and 300 IU/mL interleukin (IL)-2 (Sino Biological Inc., Chesterbrook, PA, USA, GMP-CD66).

### 2.5. Construction of Anti-EGFR CARs

ScFv1 and scFv2 were developed based on anti-EGFR antibody mAB806 [30,31]; the difference between the two was in that the former was encoded in the VH-VL orientation and the latter in the VL-VH orientation. ScFv3, scFv4, and scFv5 were derived from EGFR antibodies Y022 [32], C10 [33], and cetuximab [34], respectively, and their sequences were encoded in the VH-VL orientation. The structure of anti-EGFR CARs consisted of three parts: (1) the extracellular domain comprising human CD8a signal peptides (nucleotides 1032–1094, GenBank NM 001145873.1), scFv, and CD8a hinges (nucleotides 1443–1577, GenBank NM 001145873.1); (2) the transmembrane domain representing the transmembrane portion of CD28 (nucleotides 515–595, GenBank NM 006139.4); and (3) intracellular domains comprising CD28 (nucleotides 596–718, GenBank NM 006139.4) co-stimulated domains and CD3ζ (nucleotides 363–698, GenBank XM 011510145.2) co-stimulated polypeptides. To facilitate the assessment of the transfection efficiency in anti-EGFR CAR lentivirus-infected T cells, eGFP was linked to CD3Z via the self-cleaving peptide P2A. All anti-EGFR CAR sequences were chemically synthesized and inserted into lentiviral vectors.

### 2.6. Preparation of Anti-EGFR CAR-T Cells

Freshly enriched primary human CD3^+^ T cells were activated with Dynabeads (ThermoFisher, Waltham, MA, USA, 11131D) at a 2:1 ratio for 24 h, infected with lentivirus carrying the anti-EGFR CAR, and maintained at the density of 1 × 10^6^ cells/mL.

### 2.7. Flow Cytometry

The EGFR expression on the surface of the ESCC cell lines, anti-EGFR CAR expression in the T cells, and the proportion of human CD3^+^ T cells and anti-EGFR CAR-T cells in the mouse peripheral blood were detected by flow cytometry performed using PE-labeled anti-human EGFR (BioLegend, San Diego, CA, USA, 352904) and Brilliant Violet 421^TM^-labeled CD3 antibodies (BioLegend, San Diego, CA, USA, 300434) in CytoFLEX LX (Beckman Coulter, Brea, CA, USA) and LSRFortessa (BD) instruments.

### 2.8. Luciferase-Based Cytolysis Assay

Luciferase-expressing ESCC cells were suspended in complete RPMI 1640 medium at the density of 1 × 10^5^ cells/mL, seeded into 96-well plates (Greiner, Kremsmünster, Austria, 655098) (100 μL per well), and cultured at 5% CO_2_ and 37 °C. After 6–8 h, effector (CAR-T) cells were added (100 μL per well) at different effector to tumor (E/T) cell ratios (0.5:1, 0.25:1, or 0.125:1), and the co-cultures were incubated at the same conditions for 24 or 72 h; then, 10 μL Steady-Glo luciferase substrate (Promega, Madison, WI, USA, E2520) was added and luminescence detected using SynergyH1 (BioTek, Winooski, VT, USA). The percentage of tumor cells lysed by effector cells was calculated based on the luciferase activity: 100%—(RLU of effector and tumor cell co-culture)/(RLU of tumor cells) × 100%, where RLU indicates the relative luminescence units.

### 2.9. Cytokine ELISA

Effector cells and ESCC cell lines were co-incubated at a ratio of 0.5:1 (1 × 10^4^ tumor cells in each assay) for 3 days, and the supernatants were analyzed for the release of cytokines tumor necrosis factor (TNF)-α, IL-2, and interferon (IFN)-γ using ELISA kits (Dakewei, Shenzhen, China, 1110002, 1110202, and 1117202) according to the manufacturer’s instructions.

### 2.10. CAR-T Cell Antitumor Function in a Mouse Xenograft Model

Five-week-old female C-NKG mice (Cyagen, Santa Clara, CA, USA) were subcutaneously inoculated with 2 × 10^6^ KYSE150-luci cells. After 14 days, when the tumor volume reached 30–40 mm^3^, the mice were distributed into PBS, T, CAR1-T, CAR2-T, CAR3-T, and CAR4-T groups (*n* = 5 per group), randomly, and injected with PBS, 5 × 10^6^ T cells, or 5 × 10^6^ anti-EGFR CAR-T cells (CAR^+^ 50%) through the tail vein on days 15 and 22. The mice were monitored weekly for body weight and tumor volume; peripheral blood was collected to analyze the ratio of human cells to CAR-T cells.

### 2.11. Statistical Analysis

All of the statistical analyses for WGS and RNA-seq were performed using R (Version 4.0.2; https://www.R-project.org, accessed on 25 August 2020) and SPSS software (Version 22.0, https://www.ibm.com/analytics/spss-statistics-software, accessed on 25 August 2020). Student’s *t*-test was used to compare the expression between the tumor and normal samples. Fisher’s exact test was used to determine the association between risk scores and clinical characteristics. Kaplan–Meier curves were plotted and a log-rank test was performed. All of the subsequent experimental data were statistically analyzed using GraphPad Prism 6 and were presented as the mean ± standard deviation. Differences between groups were analyzed by Student’s *t*-test. The level of statistical significance was set at *p* < 0.05.

## 3. Results

### 3.1. EGFR Is Overexpressed in ESCC *Tissue Samples* and Cell Lines

The ESCC samples and paracancerous tissues omics data of our group were analyzed. WGS (*n* = 508) [15] analysis indicated that the EGFR copy number was significantly higher in the ESCC tissues compared to the control (Figure 1A). RNA-seq (*n* = 155) [35] differential expression analysis indicated that the EGFR expression in the tumors was strongly upregulated compared with that in the paracancerous tissues (Figure 1B). The EGFR copy number gain was significantly positively correlated with EGFR mRNA levels, suggesting that the increase in EGFR expression may be due to EGFR amplification (Figure 1C). Furthermore, the EGFR copy number gain was associated with a poor prognosis of ESCC patients (Figure 1D).

Previously, we found that ESCC patients with a high expression of EGFR were associated with a poor prognosis in a ESCC tissue microarray cohort [36]. We reanalyzed that part of the data and compared the expression level of EGFR in cancer and paracanerous tissues. We found that the expression level of EGFR in cancer tissues was significantly higher than that in paracanerous tissues (Figure 2A,B). Western blotting and flow cytometry analyses of EGFR protein levels in ESCC cell lines revealed that EGFR was overexpressed in all of them (Figure 2C,D). The WB results showed that the EGFR expression levels of KYSE150 were lower than those of KYSE450 and KYSE30, but the FACS results showed that the EGFR expression levels were similar in the three ESCC cell lines. This may be because Western blotting measures the EGFR expression levels in the total cell proteins, while flow cytometry detects the EGFR expression levels on the cell surface. All of the above results suggest that the construction of CAR-T cells targeting EGFR may have a better therapeutic effect on ESCC. Therefore, we constructed anti-EGFR CAR-T cells and tested their anti-ESCC activity. Uncropped Western blot image can be found in Appendix A.

### 3.2. Preparation and Characterization of Anti-EGFR CAR-T Cells

To construct CAR-T cells, we used a second-generation chimeric receptor structure. The variable heavy and light chain sequences of the anti-EGFR antibodies were joined through linker sequences (GGGGS GGGGS GGGGS) to generate recombinant anti-EGFR scFv, which formed the antigen-recognition portion of the anti-EGFR CAR. To provide flexibility for the CAR to recognize the target antigen, the CD8 hinger was added between the scFv and CD28 transmembrane domain (Figure 3A). Furthermore, the constructed second-generation CAR included the transmembrane domain of CD28 and the intracellular signaling domains of CD28 and CD3ζ (Figure 3A). To facilitate the assessment of the transfection efficiency of T cells with the lentivirus construct, CAR was labeled with eGFP via P2A (Figure 3A).

Five CARs targeting EGFR were constructed. The scFv portion of CAR1 and CAR2 was engineered based on the anti-EGFR antibody mAb806, which recognizes the exposed epitope EGFR^287–302^ overexpressed on the tumor cell surface, as well as mutated EGFR. The difference between CAR1 and CAR2 was that the VH of scFv1 was in the front, whereas that of scFv2 was in the back. CAR3-scFv was derived from the EGFR antibody 7B3, which also specifically binds EGFR^287–302^; CAR4-scFv was derived from low-affinity antibody C10; and CAR5-scFv represented VH-VL of cituximab, a cetuximab mutant, which is a high-affinity EGFR monoclonal antibody.

To generate anti-EGFR CAR-T cells, human CD3^+^ T cells were activated with anti-CD3/anti-CD28 Dynabeads and were infected with lentiviral vectors carrying anti-EGFR CARs. FACS analysis performed 3 days after the transfection revealed the efficiency of ~70% (Figure 3B), indicating the successful construction of EGFR-targeting CAR-T cells.

### 3.3. Anti-EGFR CAR-T Cells Are Cytotoxic for ESCC Cells In Vitro

Next, we examined the toxicity of anti-EGFR CAR-T cells for ESCC in vitro. Anti-EGFR CAR-T cells were incubated with KYSE150-luci, KYSE450-luci, or KYSE30-luci cells and the lytic activity of the effector cells was assessed based on the luciferase expression in the tumor cells. The results showed that anti-EGFR CAR1-T, CAR2-T, and CAR4-T cells had a good lytic activity against the three ESCC cell lines (Figure 4A,B). Whereas CAR3-T showed a high activity against KYSE450-luci cells, but a very low activity against KYSE150-luci and KYSE30 cells, this indicates that the toxicity of CAR-T cells to tumor cells was not only related to the targeted tumor antigen, but also had a strong relationship with the other characteristics of the tumor. CAR5-T did not show cytotoxicity for ESCC cells, so we did not use the CAR5 structure for further experiments. Co-culture of tumor and effector cells at different E/T ratios for different times indicated that anti-EGFR CAR1-T, CAR2-T, and CAR4-T cells lysed ESCC cells in a dose- and time-dependent manner (Figure 4C,D). Analysis of the co-culture supernatants for cytokine release indicated that anti-EGFR CAR1-T, CAR2-T, and CAR4-T cells secreted IL-2, IFN-γ, and TNF-α at much higher levels compared with the control T cells or anti-EGFR CAR3-T cells (Figure 5A–C). Taken together, these results show that anti-EGFR CAR1-T, CAR2-T, and CAR4-T cells were cytotoxic for ESCC cells in vitro.

### 3.4. Anti-EGFR CAR-T Cells Effectively Eliminate ESCC Tumors in a Mouse Xenograft Model

Next, we investigated whether anti-EGFR CAR-T cells could suppress the growth of ESCC tumors in a mouse xenograft model (Figure 6A). We used donor 2 to prepare CAR-T cells for in vivo experiments in mice and obtain high-efficiency CAR-T cells (Figure 6B). The results indicated that tumor volumes in the CAR1-T and CAR2-T groups were significantly reduced compared with those in the other groups, and tumors in the CAR1-T group were cleared (Figure 6C). However, the body weight of mice was not affected by CAR-T cell injection (Figure 6D). At the end of the experiment, mice were sacrificed and the weight and volume of xenograft tumors were analyzed. Consistent with previous results, mice in the CAR1-T and CAR2-T groups had smaller tumors than those in the other groups (Figure 6E–G). We also determined the proportion of human CD3^+^ T cells and anti-EGFR CAR-T cells in mouse peripheral blood using flow cytometry. The results indicated that the CAR1-T and CAR2-T groups were more abundant in CD3^+^ T cells or anti-EGFR CAR-T cells compared with the other groups (Figure 6H). Overall, these results indicated that anti-EGFR CAR1-T and CAR2-T cells had a strong cytotoxic activity against ESCC, leading to tumor elimination.

## 4. Discussion

ESCC is the predominant histological EC subtype in China; however, the effects of the existing therapeutic approaches are limited. Based on the molecular features of EGFR in ESCC, we speculated that anti-EGFR CAR-T cells might efficiently lyse ESCC cells. To test this hypothesis, we generated five anti-EGFR CAR-T cell lines, two of which showed functional activity against ESCC cells both in culture and in a mouse xenograft model. Our results provide a proof of concept that engineered anti-EGFR CAR-T cells may have a therapeutic potential for ESCC.

Molecularly targeted drugs against EGFR mainly include tyrosine kinase inhibitors (TKIs), such as gefitinib and erlotinib, and monoclonal antibodies such as cetuximab and panitumumab; however, many clinical trials have confirmed that ESCC do not respond to these drugs [37,38,39,40]. The negative clinical outcomes may be due to patients with ESCC having rare TKI-response EGFR driver mutations. Furthermore, the heterogeneity of esophageal cancer and drug resistance developed during treatment may also be responsible for the poor efficacy of molecularly targeted drugs [41].

Although ESCC patients carry few EGFR driver mutations, most of them show EGFR gene amplification and overexpression, suggesting the potential of EGFR-targeting CAR-T cells for ESCC therapy. Indeed, among the anti-EGFR CAR-T cells constructed in this study, we successfully identified two cell lines, CAR1-T and CAR2-T, with a cytotoxic activity against ESCC cells in vitro and in vivo. These effective CAR-T cell lines differ from the ineffective ones (CAR3-T, CAR4-T, and CAR5-T) only in that their scFvs are derived from other EGFR antibodies, and it can be hypothesized that the unsatisfactory results in ESCC clearance may be due to conformational variations, which may affect binding to EGFR; however, this speculation requires experimental verification. Severe toxicity, including death, has been reported due to the off-target risk of CAR T cells [42,43]. Thus, scFv selection is crucial, mAb806 binds to the EGFR^287–302^ epitope, which is exposed when EGFR overexpressed or when EGFR is mutated in cancer cells. Therefore, mAb806 can bind to EGFR overexpressed in cancer cells, but not in normal cells [30,31]. Hence, CAR1-T cells and CAR2-T cells are expected to have an excellent anti-tumor effect in esophageal squamous cell carcinoma under safe conditions. The successful construction and functional identification of anti-EGFR CAR-T cells that are cytotoxic for ESCC cells provide a preclinical basis for the application of CAR-T cell immunotherapy to treat ESCC.

Although anti-EGFR CAR-T cells are effective against non-small cell lung cancer and biliary cancer, several clinical trials suggest that CAR-T cells have a lower activity against solid tumors. One of the main reasons for this is that the solid tumor microenvironment leads to T cell exhaustion [44,45]. Therefore, the specific mechanisms underlying CAR-T cell exhaustion in the microenvironment of different solid tumors should be investigated to further improve the anti-neoplastic effect of CAR-T cells. We performed single-cell sequencing of T cells in tumor samples from patients with ESCC. An analysis of the sc-RNA-seq results is expected to assist in understanding the cause of TIL exhaustion, thereby further improving the antitumor effect of CAR-T cells in ESCC. Given the continuous replenishment of multi-omics sequencing data obtained on large ESCC samples, advances in gene editing technology, and the development of new drugs targeting diverse cancer-related immune mechanisms, it is believed that the anti-neoplastic activity of CAR-T cells against ESCC can be further improved through optimization and combination with other immunotherapies.

## 5. Conclusions

The EGFR gene was amplified and was overexpressed in ESCC tumors compared with the paracancerous tissues. Among the five constructed EGFR-targeting CAR-T cell lines, two, which expressed scFvs derived from anti-EGFR antibody mAB806, showed cytotoxicity for ESCC in vitro and in vivo. The successful construction and functional examination of anti-EGFR CAR-T cells targeting EGFR-expressing ESCC cells provide a preclinical basis for the application of CAR-T cell immunotherapy in ESCC.

## Figures and Tables

**Figure 1 cancers-14-06021-f001:**
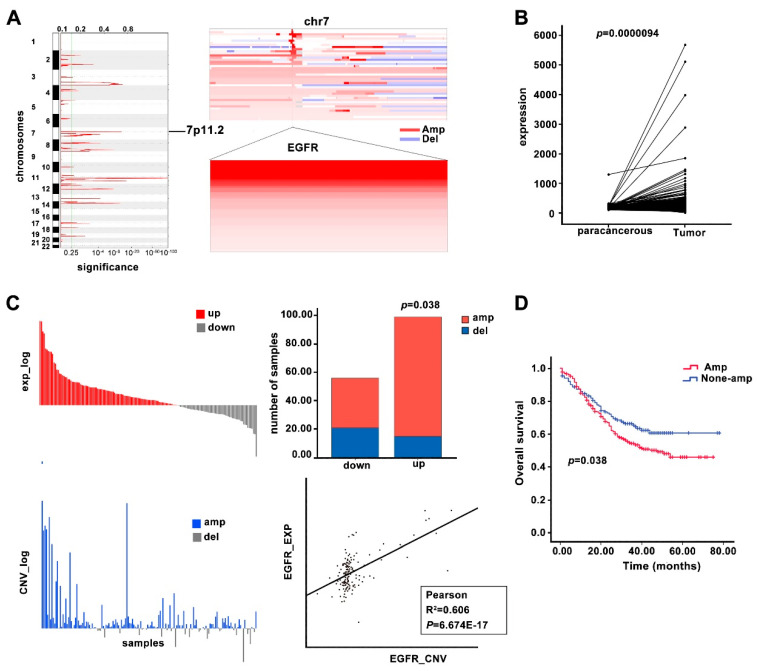
Copy number alteration and mRNA expression of the EGFR gene in ESCC. (**A**) Significant amplification of EGFR in ESCC patients (*n* = 508) revealed by WGS. Left panel: Copy number amplification region in the human chromosome of ESCC patients. Right panel: Copy number alteration of EGFR gene c in ESCC. (**B**) Differential expression of EGFR mRNA revealed by RNA-seq analysis (*n* = 155). (**C**) Correlation between EGFR amplification and expression. Top left panel: log value of expression of EGFR in each ESCC sample. Bottom left panel: log value of CNV of EGFR in each ESCC sample. Top right panel: ESCC sample number statistics of EGFR expression changes corresponding to EGFR CNV changes. Bottom right: panel: Correlation analysis of EGFR CNV and expression. (**D**) Kaplan–Meier analysis of ESCC patient survival according to EGFR gene amplification (*n* = 508). Student’s *t*-test was used to compare the expression between tumors and para-carcinoma tissues. Fisher exacts test was used to check the association of risk scores with clinical characteristics. Kaplan–Meier (KM) curves were plotted as well as a Log-rank test. Pearson was used for the correlation analysis of the EGFR amplification and expression. A *p*-value of less than 0.05 or 0.1 was set as being statistically significant.

**Figure 2 cancers-14-06021-f002:**
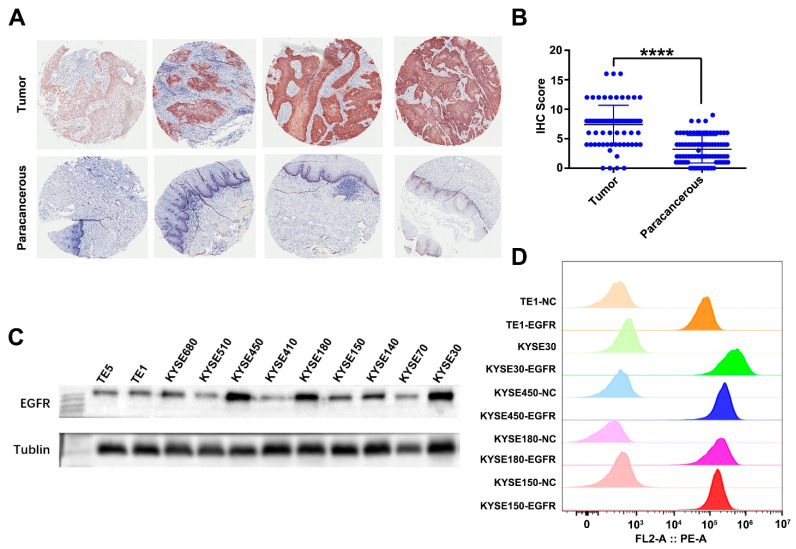
EGFR expression in ESCC tumor and paracancerous samples, and ESCC cell lines. (**A**) The expression of EGFR in ESCC tumor and paracancerou samples was detected by IHC. (**B**) The IHC score of EGFR of 85 ESCC tumor samples and paired paracancer samples. EGFR expression in ESCC cell lines TE1, KYSE30, KYSE450, KYSE180, and KYSE150 was analyzed by (**C**) Western blotting and (**D**) flow cytometry. Unpaired multiple two-tailed *t*-test. **** *p* < 0.0001.

**Figure 3 cancers-14-06021-f003:**
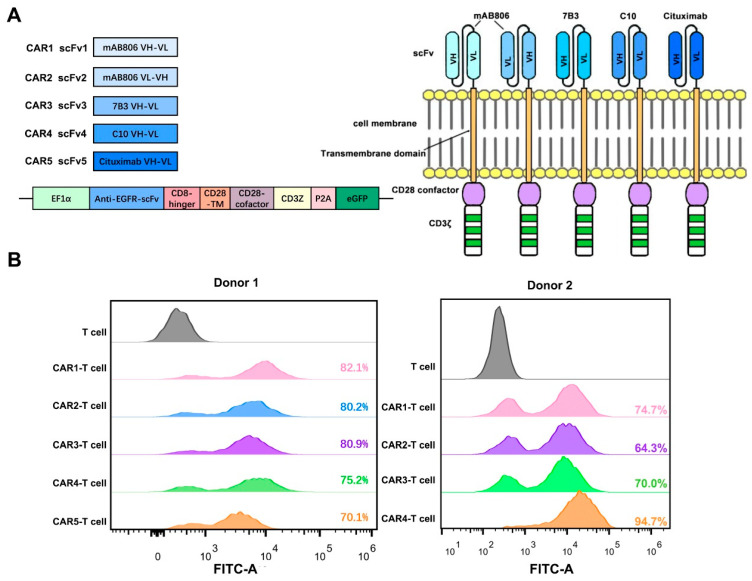
Design and expression of the anti-EGFR CAR. (**A**) Schematic illustration of the five second-generation anti-EGFR CAR constructs. (**Left**) Plasmid structure schematic, (**Right**) CAR structure schematic. (**B**) Expression levels of anti-EGFR CARs in T cells according to FACS analysis. T cells were isolated from PBMCs of two different donors.

**Figure 4 cancers-14-06021-f004:**
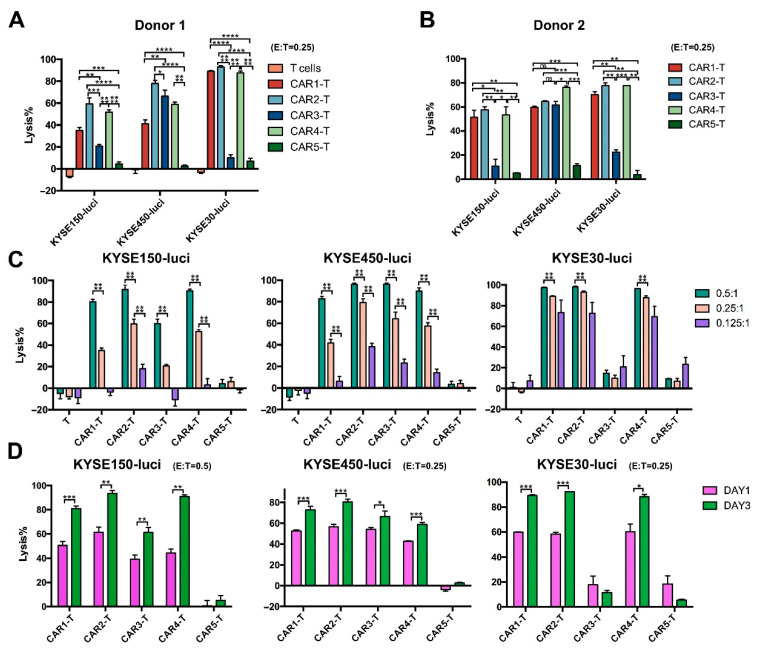
Antitumor effects of five anti-EGFR CAR-T cell lines in vitro. Anti-EGFR CAR-T cells obtained through lentiviral vector transfection were incubated with ESCC cell lines (KYSE150, KYSE450, or KYSE30) at different effector to target (E/T) ratios for different times and were analyzed for cytotoxicity using a luciferase-based cytolysis assay. (**A**,**B**) Incubation at the E/T ratio of 0.25:1 for 3 days; (**C**) incubation at the E/T ratios of 0.5:1, 0.25:1, and 0.125:1 for 3 days; (**D**) incubation at the E/T ratio of 0.5:1 or 0.25:1 for 1 or 3 days. Unpaired multiple two-tailed *t*-test. * *p* < 0.05, ** *p* < 0.01, *** *p* < 0.001, and **** *p* < 0.0001.

**Figure 5 cancers-14-06021-f005:**
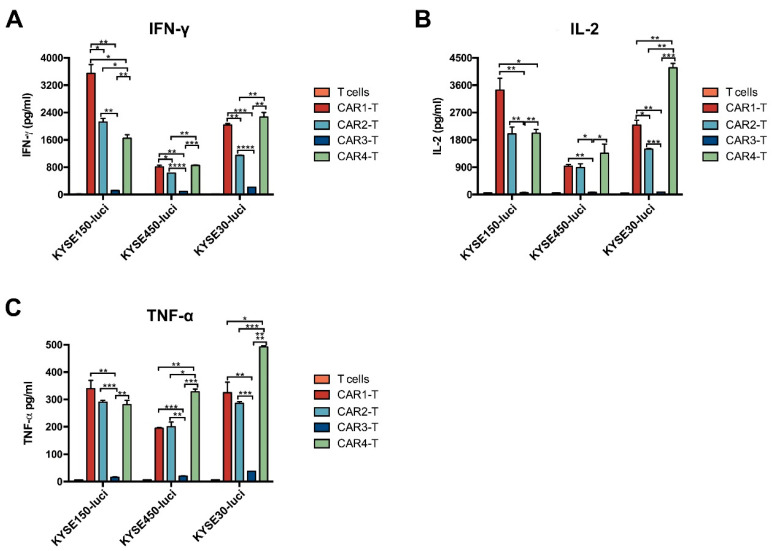
Secretion of IFN-γ, IL-2, and TNF-α by anti-EGFR CAR-T cells. Anti-EGFR CAR-T cells were incubated with KYSE450, KYSE150, and KYSE30 cells for 3 days, and culture supernatants were analyzed for the concentrations of (**A**) IFN-γ, (**B**) IL-2, and (**C**) TNF-α. Unpaired multiple two-tailed *t*-test, * *p* < 0.05, ** *p* < 0.01, *** *p* < 0.001, and **** *p* < 0.0001. EGFR, epidermal growth factor receptor; IFN-γ, interferon γ; IL-2, interleukin 2; TNF-α, tumor necrosis factor α.

**Figure 6 cancers-14-06021-f006:**
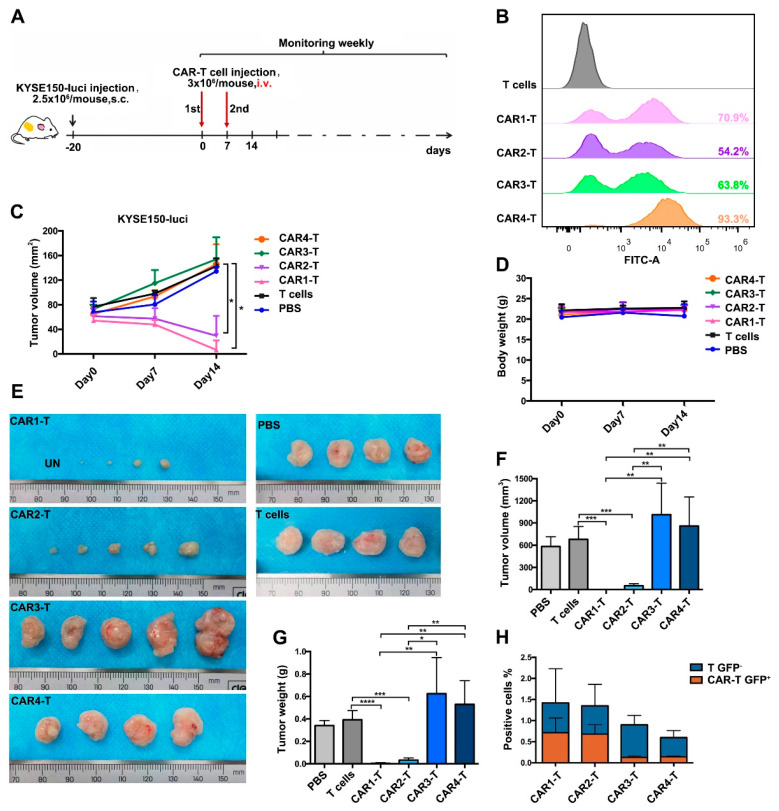
Antitumor effects of anti-EGFR CAR-T cells in vivo. (**A**) Experimental design of the in vivo functional test for CAR-T cells. (**B**) Expression of the anti-EGFR CAR in T cells used for the in vivo experiment was analyzed by FACS. (**C**) Tumor volume changes after intravenous injection of anti-EGFR CAR-T cells into CDX mice with KYSE150-luci cell xenografts. (**D**) Mouse body weight after anti-EGFR CAR-T cell administration. (**E**–**G**) Tumor sizes (**E**,**F**) and weights (**G**) at the end of the experiment. (**H**) The proportion of hCD3+ cells and CAR-T cells in the peripheral blood of mice was analyzed before the mice were sacrificed. Unpaired multiple two-tailed *t*-test, * *p* < 0.05, ** *p* < 0.01, *** *p* < 0.001, and **** *p* < 0.0001. CDX, cell-line-derived xenograft; i.v., intravenous injection.

## Data Availability

The WGS and RNA-seq data of ESCC used in this paper can be found in other published work of our group [15,35], and the raw sequencing data generated in this study have been deposited in the Genome Sequence Archive, China National Center for Bioinformation/Beijing Institute of Genomics, Chinese Academy of Sciences (GSA-Human): HRA003107 (WGS&RNA-seq, https://ngdc.cncb.ac.cn/gsa-human/browse/HRA003107) and HRA000021 (WGS, https://ngdc.cncb.ac.cn/gsa-human/browse/HRA000021).

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
