# Peer review of "Role of Epidermal Growth Factor Receptor-Specific CAR-T Cells in the Suppression of Esophageal Squamous Cell Carcinoma"

_cancers, 2022, doi:10.3390/cancers14246021_

Round 1
Reviewer 1 Report
Overview and general recommendation:
In the research, the authors employed WGS and RNA-seq to establish that EGFR is overexpressed in ESCC tissues than in paracancerous tissues. Then they construct Anti-EGFR CAR-T Cells and following in vitro data show that Anti-EGFR CAR1-T, Anti-EGFR CAR2-T and Anti-EGFR CAR4-T are cytotoxic for ESCC cells and can cause inflammatory response. Their in vivo assay further show that Anti-EGFR CAR1-T and Anti-EGFR CAR2-T can eliminate ESCC Tumors.
I find the paper is organized in a proper way. The authors perform background research carefully. Major methods are well described in the manuscript and properly used in the research. The result is good enough to support their conclusions. The figures are well organized and presented in an appropriate way. However, I still think there is something can be improved. I suggest the authors to include more information about how to design the experiment.
Major comments:
1. I noticed that T cells isolated form two donors are used in the experiment. In figure 4A, Tell carrying no plasmid serves as a control but in figure 4B, there is no such control. Can the authors explain why the setup of the experiment is different with T cells isolated from 2 different donors?
2. How the authors determine the ratio when co-culturing EGFR CAR-T cells with tumer cells to examine the toxicity of anti-EGFR CAR-T cells for ESCC in vitro? (Fig4A, 4B)
3. In figure4C, it seems that for KYSE150-luci, KYSE450-luci and KYSE300-luci, when the E:T=0.5:1, the anti-EGFR CAR-T cells show strongest effect. But in figure4D, when work with KYSE150-luci, the ratio is E:T=0.5:1. When work with KYSE450-luci and KYSE300-luci, the ratio is E:T=0.25:1. Can the authors explain why they use different ratio for different cells in the same experiment?
4. In discussion part, I suggest the authors to include more references and information to emphasize the importance of this research.
Minor comments:
1. Figure 6B is not referenced in the manuscript.
Reviewer 2 Report
The authors have explored the role of Epidermal Growth Factor Receptor-Specific CAR-T Cells in the Suppression of Esophageal Squamous Cell Carcinoma. Initially, they checked the expression level of EGFR in Esophageal Squamous Cell Carcinoma patients and then they tried to prepare car T cells and showed their anticancer role. Here is my observations that I would like to share.
1. Figure 1 is not clearly presented. I could not see in the legend what figure represent what experiment and there is no labelling on the axis of any figures.
2. For example in Figure 1A what are the numbers on left side and what are the numbers in the bottom? Please clarify in all the figures. And also mention in figure legend
3. In Figure 2B author should calculate the band intensity and plot it with a significance test.
4. Author did not discuss their results in detail. More discussion is needed.
The manuscript require major revision.
Reviewer 3 Report
1. The authors didn’t describe the data source of Fig1, are they original data from this research?
2. In Fig2, could the authors show the EGFR expression level in non-tumor tissues?
3. In Fig2 A, KYSE150 expresses as high EGFR as KYSE 180 and KYSE450, but Fig2B shows KYSE 150 expresses less EGFP than those two cell lines.
4. Based on Fig 4, CAR2 worked better than CAR1 in Dnor1 after co-culturing with KYSE150, however, CAR1 exhibited increased cytokines production in Fig5. It’s hard to connect these two phenotypes. Could the authors show more evidence, such as GzmB, Prf1 production, or CD107 internalization? Did CAR2 upregulate REAL cytotoxicity?
5. Due to the different in vitro CTL efficiency of Donor 1 CARs and Donor 2 CARs, I wonder which cells were used in the in vivo experiments.
6. Fig6H showed the proportions in the PBMC, did the CAR3 and CAR4 lose GFP expression or show less infiltration in the tumors?
Reviewer 4 Report
The authors show in this manuscript data with 5 second generation chimeric constructs of CAR-T cells targeting EGFR with the goal to treat Esophageal Squamous Cell Carcinoma. Two of them with activity in vitro and in vivo have a potential to be used for treating ESCC.
This manuscript is interesting but need complements:
1/ Regarding the constructs:
A figure with schematic presentation of the different constructs should be added at the beginning of the manuscript and more readable than as shown in Figure 3A.
More generally, as the authors mentioned in Introduction, other CAR-T cells targeting EGFR are published and under development. What are the pros and cons of constructs presented here in comparison to the already existing anti-EGFR CAR-Ts? Nothing is said. This point has to be developed at least in Discussion to really evaluate the advantage(s) or not of these new constructs.
2/ Regarding WGS:
Source of tissue samples is not mentioned.
No reference to patient consent as well.
WGS method is not described.
Figure 1 should be enlarged to be readable.
3/Regarding EGFR detection:
Both techniques confirm EGFR expression but the 2 techniques did not show comparable expression. Indeed, for example, flow cytometry show a quite similar level of expression with KYSE150 and KYSE450 cells, but the Western-blot shows clearly a higher expression in KYSE450 cells. These differences are not commented by the authors.
Kits for quantifying of receptor expression by facs exist, that could lead to a more precise evaluation and that should permit to compare later cytotoxicity and receptor rexpression.
IHC images should be presented at list as supplementary data to visualize EGFR overexpression in ESCC tissues.
4/ Regarding cytotoxicity:
These data could be pore comment regarding the level of expression of EGF-R in cell lines.
It is interesting to notice that CAR5-T is inactive whilst it is the only CAR-T performed based on an anti-EGFR antibody in clinic. Have the authors an explanation of this observation ?
CAR3-T is not active on all cell lines tested. Why such a selectivity ? Is it due to level of over-expression of EGF-R ? or due to expression of other receptors partner of EGF-R ?
5/ Regarding cytokines secretion:
Why CAR5-T is not presented here? That could be a good negative control.
6/ Regarding the in vivo experiments:
Body weight loss was not observed but what are the epitopes of scFv tested ? If they do not cross-react with murine receptor, an lack of sign of toxicity is not surprising an this indirect evaluation of toxicity is not significant.
Round 2
Reviewer 2 Report
In Figure 2B (Western blot), the protein loading was not equal as observed by the GAPDH band. the intensity of the GAPDH band is different in all the samples. Please reperform the western with equal protein loading.
Reviewer 3 Report
The authors have addressed most of my questions.
Author Response
Response: Thank you very much for your precious time, effort and support on this work. We appreciate your comments and suggestions.
Reviewer 4 Report
The authors replied to most of the questions but some comments need to be added in the publication:
Regarding the strategy versus the previous CAR-T cells, reply should not be restricted to the reviewer and added in Introduction.
Regarding the detection of EGFR, I totally agree the comment. Therefore, I suggest to specify in the text the detection of membranous expression by flow cytometry and global expression by Western-blot.
Regarding the cytotoxicity, some comments should be added in the text as well because any reader could have the same questions.
Finally, the authors replied concerning the epitope and the selectivity tumor vs normal cells but they did not reply concerning the cross-reactivity (or not) with murine EGFR. This point has to be clarified for evaluating the signification of body-weight loss. Obviously, if there is no cross-reactivity, toxicity cannot be really evaluate on this model (and that has to be mentioned).
